# LEVERAGE UNLEARNING TO SANITIZE LLMS

## ABSTRACT

Pre-trained large language models (LLMs) are becoming useful for various tasks. To improve their performance on certain tasks, it is necessary to fine-tune them on specific data corpora (e.g., medical reports, business data). These specialized data corpora may contain sensitive data (e.g., personal or confidential data) that will be memorized by the model and likely to be regurgitated during its subsequent use. This memorization of sensitive information by the model poses a significant privacy or confidentiality issue. To remove this memorization and sanitize the model without requiring costly additional fine-tuning on a secured data corpus, we propose SANI. SANI is an unlearning approach to sanitize language models. It relies on both an erasure and repair phases that 1) reset certain neurons in the last layers of the model to disrupt the memorization of fine-grained information, and then 2) fine-tune the model while avoiding memorizing sensitive information. We comprehensively evaluate SANI to sanitize both a model fine-tuned and specialized with medical data by removing directly and indirectly identifiers from the memorization of the model, and a standard pre-trained model by removing specific terms defined as confidential information from the model. Results show that with only few additional epochs of unlearning, the model is sanitized and the number of regurgitations is drastically reduced. This approach can be particularly useful for hospitals or other industries that have already spent significant resources training models on large datasets and wish to sanitize them before sharing.

## 1 INTRODUCTION

Recent advances in NLP Singh & Mahmood (2021) based on neural networks have democratized their use Chen et al. (2022c); Yeo et al. (2022); Wei et al. (2022); Goyal et al. (2024). Since the advent of ChatGPT, NLP models are not limited to text generation and can include several downstream tasks, such as classification tasks Zhang et al. (2024) or Named Entity Recognition (NER) Zhao et al. (2024).Pre-trained large language models (LLMs) such as BERT or GPT have significantly contributed to their adoption across all sectors. These off-the-shelf models were pre-trained at great expense on countless unlabeled datasets web-scraped from the Internet. These pre-trained foundation models can then be fine-tuned on specific datasets and even trained for new tasks by adding a few additional layers. For example, a hospital will fine-tune a BERT model by leveraging its medical reports before training it with an additional layer for a downstream classification task.

The need for information sharing extends beyond data and now concerns learning models. For example, some hospitals also want to share specialized models fine-tuned on their own data with other medical centers.

The attack surface on models is still poorly understood (Weidinger et al. (2022); Lehman et al. (2021); Duprieu & Berkouk (2024)). A number of threats are related to the memorization and possible leakage of sensitive information used during model training, such as data reconstruction and membership inference (i.e., identifying elements used during the training or the fine-tuning). For example, the European Data Protection Board (EDPB) issued an opinion on the development and deployment of AI models in light of the GDPR [1]. Specifically, the opinion addresses the issue of AI models trained with personal data and stipulates that to be considered anonymous, the likelihood

---

[1] https://www.edpb.europa.eu/news/news/2024/edpb-opinion-ai-models-gdpr-principles-support-r
en

of regurgitation of personal information must be insignificant, taking into account all the means reasonably likely to be used.

Therefore, it is necessary to ensure that the model is sanitized from any sensitive information memorized before sharing it. This sensitive information may be personal information or other confidential information (e.g., related to intellectual property). Since training or specializing a model requires significant computational resources and is time-consuming, retraining or re-specializing from scratch would be prohibitively expensive. To overcome this limitation, we develop SANI, a system that sanitizes LLMs by removing sensitive information from the memorization of the model. SANI leverages a unlearning scheme to remove specific information (i.e., information present in a blacklist) from the model at a reduced cost. Specifically, SANI adapts an erase and repair strategy (Kurmanji et al. (2023b)) where the former phase reinitializes some neurons in the last layer (i.e., capturing and disrupting only fine-grained memorization) without affecting the model foundations memorized in the lower layers of the architecture, while the latter phase restores and specializes models while avoiding the memorization of terms defined in a blacklist (Boutet et al. (2025)).

We conducted an empirical evaluation of SANI by measuring its ability to unlearn personal information memorized in a pre-trained LLMs (i.e., BERT and GPT models) fine-tuned with medical data, and to unlearn information defined as confidential exposed to a LLM during its training. We demonstrate that SANI is able to quickly remove sensitive information from the model without altering its performance for word prediction or generation, as well as for downstream classification tasks. Furthermore, we show that the sensitive information the most exposed to the model is the more affected by the unlearning, drastically reducing the number of regurgitations. Finally, compared against different baselines, we show that SANI offers the best trade-off between reducing sensitive information regurgitation and model performance.

## 2  BACKGROUND AND RELATED WORK

Large language models (LLMs) are trained on very large datasets. For example, training chatGPT is very costly and required years of crawling the Internet. LLMs (i.e., foundation models) are generally optimized for specific tasks with fine-tuning using domain-specific data. In the medical field, for example, these models are specialized with patient reports, which improves performance for downstream tasks (Huang et al. (2020)). Without specific measures, these models can therefore memorize and subsequently regurgitate and disclose potentially sensitive information from the training data, which constitutes a significant privacy leak and has implications for the practical and legal aspects of these models. The same problem arises with a model exposed to confidential data during its training, and which could then be led to regurgitate it during its exploitation. To avoid training models once again from scratch (by excluding sensitive information from the beginning), unlearning aims to forget or removing specific data (e.g., sensitive, unreliable, or copyrighted data) from the memorisation of the model. This section presents a comprehensive background and related work on unlearning (Section 2.1), and on the risks of memorization of training data by the models (Section 2.2).

### 2.1  MACHINE UNLEARNING

Machine Unlearning (Cao & Yang (2015); Xu et al. (2023); Nguyen et al. (2024); Kurmanji et al. (2023b)) has rapidly emerged as a significant area of research within the field of Machine Learning (ML). A first approach (Bourtoule et al. (2020)) introduces a framework that utilizes data shading and slicing techniques to minimize the computational overhead associated with unlearning. Data shading involves modifying data representations to obscure sensitive information, while slicing refers to dividing data into manageable segments to simplify the unlearning process that completely remove the influence of specific data from a ML model. Similarly, Chen et al. (2022a) applies a comparable strategy to recommendation systems, advocating the creation of multiple sub-samples to reduce the impact of forgetting particular observations. Although these methods can help manage the computational load, they may still be resource-intensive if a large amount of data needs to be forgotten.

To address these challenges, more recent research has focused on developing efficient unlearning techniques that aim to approximate a full retraining from scratch while minimizing the associated costs. For instance, Kurmanji et al. (2023a) introduces the SCRUB method, which frames unlearn-

ing as a teacher-student problem. This approach tackles various tasks associated with unlearning, such as removing bias, resolving confusion, and safeguarding user privacy. Another method, proposed by Graves et al. (2020), involves maintaining a record of which examples appeared in which training batches and tracking the corresponding parameter updates. By undoing updates related to sensitive information, this method effectively removes the influence of such data from the model. Following a recent challenge on unlearning, Triantafillou et al. (2024) analyzed progress in the field and highlighted strategies based on erasure and repair phases among the best solutions. Finally, often used to apply the right to be forgotten to the learning model, unlearning strategies have also been applied for backdoor removal (Min et al. (2025)).

Regarding machine unlearning in LLMs, previous approaches used gradient ascent (GA) over undesired knowledge to inversely optimize models (Jang et al. (2022); Yao et al. (2024)) Other unlearning schemes for LLMs are based on more and less selective pruning strategies (Pochinkov & Schoots (2024); Liu et al. (2025); Zhang et al. (2025). For instance, Chen et al. (2024) propose a solution to localize neurones that memorize sensitive informations in order to erase only these neurones. SANI also takes advantage of this two-phase unlearning strategy with targeted erasure and controlled reconstruction.

## 2.2 RISK OF MEMORIZATION OF TRAINING DATA

A central question related to the risks of memorization of LLMs concerns the extent to which models memorize their training data (Carlini et al. (2019; 2023); Nasr et al. (2023); Zhang et al. (2021); Schwarzschild et al. (2024)). We can notably cite extractable memorization and membership inference.

**Extractable memorization:** Extractable memorization is a type of attack that aims to use the model to infer information from the original data (Liu et al. (2021)). This attack mainly concerns text generation models, such as GPT. These models are trained to produce text based on what they have seen during training. However, the model is not expected to be a basic parrot and repeat exactly the sentences it has seen. This is especially concerning if the data it is repeating is sensitive. This has been shown to be the case with GPT-2 for example, from which the names and addresses of individuals can be extracted Carlini et al. (2020). Compressible memorization (Schwarzschild et al. (2024)) extends this definition by evaluating how short the minimal requested sentence (or prompt) that elicits the sequence.

**Membership inference:** Membership inference attack (MIA) is a more common inference attack in machine learning, which aims to infer whether a specific data was used in the training data of a target model (Carlini et al. (2022)). There are different techniques that can be used to perform a MIA attack depending on if the adversary has an access to the model parameters (i.e., white-box access), or access to a ground-truth subset of member and non-member samples. One technique consists to analyze the loss of member and non member samples (Yeom et al. (2018)), another one is to use multiple shadow models (Shokri et al. (2017); Ye et al. (2022)) trained to mimic the behavior of the target model on an auxiliary dataset. An adversarial model is then trained to infer membership from the loss or from shadow models. Another method (Zhang et al. (2021)) is based on comparing the performance of the target model trained on a dataset with a specific input, with a second model trained without it. As ML models are supposed to learn general information, one piece of data (even rare, outlier or mislabelled samples) is not supposed to be memorized and significantly changed the model's performance. By repeating this operation many times with different subset, it is possible to identify counter-factually memorized data.

**Mitigation strategies:** The most popular approach to mitigating memorization is Differential Privacy Dwork (2006) (DP). DP is a mathematical property that a model must satisfy in order to disclose as little information as possible. This property requires the model to learn a limited amount of information at each training step. The most popular method to apply DP in machine learning is DP-SGD: Differentially-Private Stochastic Gradient Descent (Abadi et al. (2016)). The idea is to apply DP during the training phase by clipping gradient updates and adding noise at each step. DP is known to significantly decrease the accuracy of the model (Jagannatha et al. (2021)) and privacy budget management is difficult.

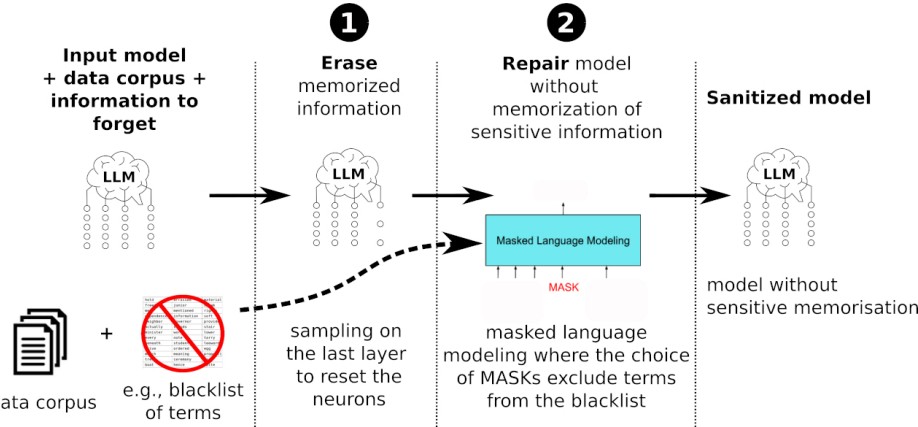

Figure 1: Overview: once the information to be unlearned is defined, SANI relies on erase and repair phases allowing to disturb only the memorization of fine information before repairing the model by specializing it while ensuring the absence of memorization of the terms to be sanitized.

To reduce privacy risks, Chen et al. (2022b) proposed to relax the loss in order to reduce the distinguishability between the training and testing loss distributions, and Chen et al. (2024) proposed a method to localize specific neurons responsible for memorizing PII in LLMs through adversarial training and then disable these neurons. To avoid memorizing personal information used during model specialization, Boutet et al. (2025) proposes a language modeling scheme that first identifies the direct and indirect identifiers, and secondly applies a language modeling scheme that avoids choosing these identifiers as masks. We leverage this language modeling scheme in our second unlearning phase to repair the model without memorizing terms defined as sensitive.

## 3 SANI: SANITIZING LLMS

In order to sanitize LLMs, we have developed SANI. SANI relies on an unlearning scheme to remove specific pieces of information (e.g., sensitive, unreliable, or copyrighted data) from the memorisation of the model. Figure 1 depicts an overview of how SANI works and the steps it involves. It relies on the identification of sensitive information to forget (Section 3.1), and two steps, a selective erasure (Section 3.2) and a repair without without memorisation of sensitive information (Section 2.2).

### 3.1 SENSITIVE INFORMATION TO FORGET

First, SANI needs as input the LLM to be sanitized and the identification of the sensitive information to forget and unlearn from the model. Depending on the purpose of the sanitization, the nature of the information to be forgotten can be different. For example, if the goal is to anonymize the model, the sensitive information to be forgotten concerns all direct and indirect identifiers linked to individuals. If, on the other hand, the goal is to share a model trained with business data, the sensitive information to be forgotten will be all confidential or copyrighted information that could have been exposed and memorized by the model. Another use case could be to remove biases from the model. In this latter case, the sensitive information to be forgotten would be outliers, aberrant data or other poisoned data. This sensitive information to forget consists of a list of single or several words (i.e., n-grams) contained in the training data.

### 3.2 TARGETED ERASURE

Once the LLM and the sensitive information to forget are defined, SANI conducts a targeted erasure. Specifically, the erasure step (step ❶ in the figure) resets certain neurons in the model's architecture to disrupt its memorization. To control this reset and reduce its detrimental effect on memorization, only the last layer of the model is affected, with a random reset of 50% of the neurons. By targeting erasure only on the last layer, fine-grained memorisation is disrupted without affecting the model's

learning foundation Jourdan et al. (2021). This allows, among other things, to maintain strong model convergence dynamics during subsequent learning steps, carried out during the reconstruction phase. An LLM (e.g., a BERT model) is built from a series of attention heads, followed by a linear layer. The purpose of the attention heads is to capture dependency information between words. The last linear layer uses this dependency information for a specific task, such as text or token classification. It is in this last linear layer that the weights are reset, which does not affect the memorization of the dependency information between words of the other layers.

### 3.3 REPAIR WITHOUT SENSITIVE MEMORISATION

In the last reconstruction step (step ❷ in the figure), controlled training cycles are performed. Specifically, the model is specialized through language modeling applied to the model's training data, while ensuring that sensitive blacklisted terms are not memorized by the model Boutet et al. (2025). To achieve this, the masks defined during this specialization phase are randomly selected from the input data set (i.e., the model's training data), while excluding terms contained in the blacklist (these terms can, however, serve as context for predicting another mask). Ensuring that these terms are not memorized in the model through the language modeling (i.e., the masked language modeling for BERT-like models, and the causal language modeling for GPT-like models) reduces the model's ability to regurgitate them (in a generative model like GPT, for example) or predict them (in a predictive model like BERT, for example). Since only the last layer of the model was modified during the erasure phase, this fine-tuning performed in the reconstruction phase converges quickly, and only a limited number of epochs are required to converge to a utility close to the original model.

## 4 EVALUATION

We performed an extensive evaluation of SANI on two realistic use cases using medical and book datasets, and we considered both off-the-shelf BERT and GPT pre-trained models. To capture the full impact of our proposed solutions, we considered a set of metrics capturing both the utility and the risk of regurgitation of sensitive information and we compared SANI against two baseline approaches. The main results show that SANI unlearns very quickly sensitive information and that the number of regurgitations decreases drastically after only 1 epoch. The most repeated sensitive terms in training (or in the specialization) are the terms most affected by unlearning.

### 4.1 EXPERIMENTAL SETUP

This section provides details about our experimental setup. Additional methodological information to replicate our evaluations is provided in the Appendix A.1.

#### 4.1.1 USE CASES

To evaluate SANI we considered two realistic use cases: to sanitize fine-tuned models, and to sanitize off-the-shelf pre-trained models.

**Sanitizing fine-tuned models:** We consider here the example of a hospital that has already spent resources to fine-tune a pre-trained model on its own medical records and wishes to anonymize this model without having to conduct costly training and specialization again. To implement this use case, we consider both a pre-trained BERT and GPT models fine-tuned on raw medical records and a models fine-tuned on pseudonymized medical records. The goal of the sanitization is to ensure that all identifiers referring to an individual are not regurgitated by the models. This includes unlearning direct and indirect identifiers in the case of a specialized model with raw data, and, respectively, unlearning indirect identifiers in the case of a specialized model with pseudonymized data.

**Sanitizing pre-trained models:** We consider here a company that would like to ensure that a model it has trained with its own data does not contain confidential information. To implement this use case, we consider a pre-trained BERT model (i.e., trained using a very large amount of web-scraped data) and the goal of the sanitization is to unlearn certain terms categorized as confidential that were used in texts known to be part of the model's training. More specifically, we identified terms belonging to specific categories (identified via a NER model) contained in a public dataset used when training the BERT model, and the goal is to ensure that the model does not regurgitate them.

In both cases, to justify unlearning, it must be significantly less expensive than completely retraining the model by excluding the data to be removed (Muresanu et al. (2025)). A 20% barrier has been established, beyond which unlearning is considered too costly (Triantafillou et al. (2024)).

### 4.1.2 DATASET

To conduct our experiments on health data (the use case on sanitizing fine-tuned models), we considered the N2C2 datasets (Henry et al. (2020)). More precisely, we leverage a dataset with medical discharge summaries (Uzuner et al. (2007)) gathering 928 records (i.e., English free text) that have been annotated for replacing all authentic Personally Identifiable Information (PII) with realistic surrogates. To identify sensitive information used to pre-train BERT model (the use case on sanitizing pre-trained models), we used the BookCorpus dataset (Zhu et al. (2015)), known to be part of the training of BERT model (Devlin et al. (2019)). BookCorpus is a popular large-scale dataset consisting of the text of around 7,000 self-published books scraped from the indie ebook distribution website Smashwords. The dataset consists of around 985 million words, and the books that comprise it span a range of genres. We only use the first 500,000 texts from the dataset available on Hugging-Face[2], which represents 77 million words. To assess the impact of unlearning on a classification downstream task, we trained a model to classify emotions associated with a text. To achieve that, we used a dataset gathering Twitter messages (Saravia et al. (2018)). The tweets were annotated and spanned into eight basic emotion categories: anger, anticipation, disgust, fear, joy, sadness, surprise, and trust.

### 4.1.3 MODELS

To cover both descriptive and generative uses of language models, we considered both the BERT and GPT architectures (Radford et al. (2019)). For BERT, we considered the base model composed of 12 layers, 768 hidden dimensions, 12 attention heads, and 110 million parameters, and for GPT, we considered the smaller version of GPT-2 with 124 million parameters, both from Hugging Face. We did not consider larger versions of the model due to limited resources, and we also discarded the distilled version of these models.

We also leveraged a off-the-shell NER (Named Entity Recognition) model from HugginFace[3] used to identify terms categorized as confidential in the use case on sanitization of pre-trained model. Specifically, this model is a bert-base-cased model that was fine-tuned on the English version of the standard CoNLL-2003 Named Entity Recognition dataset (Tjong Kim Sang & De Meulder (2003)). Terms classified as PLACE, PERSON, ORGANIZATION, and MISCELLANEOUS are marked as confidential in our use case and must be unlearned. This includes 380 different words and represents 1,953,000 iterations including repetitions.

### 4.1.4 METRICS

We consider different metrics in our evaluation to better capture the trade-off between the quality of the unlearning and the performance of the model.

**Utility:** For the performance evaluation of BERT-like models, we quantify the ability of the models to predict masked terms well (measured by the prediction accuracy). For GPT-like models, this performance is evaluated by the model's ability to generate the right tokens in the right places (measured by the tokens accuracy). To evaluate the performance of the pre-trained model, we considered a downstream classification task. More specifically, we measure the model's performance through an emotion classification task (measured by the F1-Score metric on emotion prediction).

**Regurgitation:** To assess the risk of regurgitation of terms that should have been unlearned, we study the model's ability to predict both identifiers (direct and indirect) and terms marked as confidential, contained in the dataset used to fine-tune or train it. As soon as one of these terms is predicted, whether in the correct place in the original text or elsewhere, we count a regurgitation. The metric named *Privacy* measures the ability of a model to predict any identifying term (i.e., direct or indirect identifier). A Privacy=1 means that no identifiers are predicted or regurgitated by the model, and a Privacy=0 means that all iterations of identifiers are regurgitated. The metric named

---

[2]Hugging Face - https://huggingface.co/
[3]bert-base-NER: https://huggingface.co/dslim/bert-base-NER

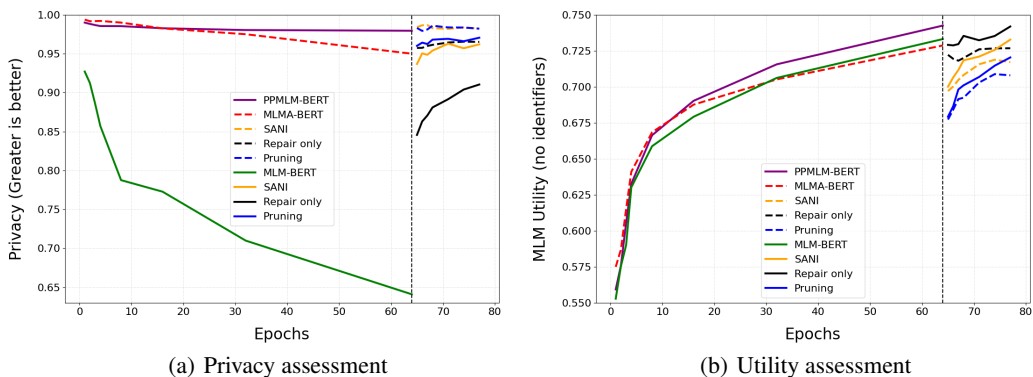

Figure 2: Sanitizing a fine-tuned BERT model: a single epoch is enough to drastically reduce identifier regurgitation while limiting utility reduction.

*Regurgitation*, in turn, quantifies the ability of the model to predict a term marked as confidential. A Regurgitation=1 means that all iterations of confidential terms are predicted by the model, and conversely, a Regurgitation=0 means that none are predicted.

#### 4.1.5 Comparative Baselines

We compare our unlearning scheme SANI against different comparative baselines.

**Repair only:** Unlike new unlearning schemes that include an erasure and a repair phase, this baseline includes only a repair phase. This repair phase is performed under the same conditions as SANI, i.e., the masking performed in the model specialization operation excludes the terms to be unlearned.

**Pruning:** This baseline consists of an erasure and a repair phase. In the erasure phase, the neurons in each layer of the model are sorted in ascending order according to their weight, the 1% of the most relevant neurons (i.e., with the highest weights) remain unaffected, while 20% of the remaining 99% are randomly reset. The repair phase is identical to SANI and the repair-only baseline.

**PPMLM-BERT and PPCLM-GPT**: This baseline (which does not implement unlearning) corresponds to a complete new specialization of the model that avoids memorizing the terms to be unlearned from the beginning. This baseline implements the solution proposed by Boutet et al. (2025), and represents the lower bound in terms of regurgitation.

### 4.2 Sanitizing a fine-tuned model

Here, we evaluate the ability of SANI to sanitize the identifiers of a Masked Language Model (i.e., a BERT model) specialized with medical data. Figure 2 depicts the privacy and utility trade-off of the BERT model during both its fine-tuning (i.e., until epoch 64) and during its sanitization (i.e., from epochs 64 to 78). During fine-tuning, we distinguish three curves: one corresponding to a specialization without protection scheme (named MLM-BERT), a specialization from a pseudonimized dataset where the directly identifiers has been replaced by "X" (named MLMA-BERT), and a specialization that has integrated a protection mechanism to avoid memorizing both direct and indirect identifiers (named PPMLM-BERT). First, the evaluation of the privacy (Figure 5(a)) shows that without protection scheme (i.e., MLM-BERT), privacy drops during the specialization, the more the number of epochs increases, the more privacy decreases. This drop in privacy shows that the model tends to memorize more and more identifiers over the course of specialization. Next, we observe that the privacy of MLMA-BERT (corresponding to a specialization using pseudonymized data, i.e., without direct identifiers) also decreases over the course of specialization, but significantly less. This is due to the fact that the model only memorizes indirect identifiers, which are less abundant than direct identifiers. Finally, by avoiding the memorization of identifiers (both direct and indirect), PPMLM-BERT maintains approximately the same level of privacy as specialization progresses.

Here we consider two models to be sanitized. The first is the model fine-tuned without protection (i.e., MLM-BERT at epoch 64) and the goal is to sanitize the model from all identifiers (direct and indirect) memorized by the model. The second is the model fine-tuned using pseudonymized data (i.e., MLMA-BERT at epoch 64) and the objective is to sanitize this model from all indirect identifiers memorized by the model (the direct identifiers are already absent in the data used to fine-tune the model). The results show that in both cases, privacy increases very rapidly. For example, a single epoch is enough to increase the privacy of a specialized model without protection by approximately 0.2, 0.3, and 0.32 using an unlearning strategy based on repair only, SANI, and pruning, respectively. The privacy of a specialized model with pseudonymized data also increases to a lesser extent after one epoch. In both cases, privacy continues to progress until the end of the unlearning cycle at epoch 77 (which corresponds to 20% of the cost of the specialization). At epoch 77, all unlearning strategies except repair-only, display a privacy level close to that of a privacy-preserving specialization solution that does not memorize identifiers (i.e., privacy close to 1, the upper bound).

Figure 5(b), in turn, depicts the utility assessment during the specialization (until epoch 64) and during the sanitization (from 65 to 77). During specialization, all approaches (i.e., without protection MLM-BERT, using pseudonymized data MLMA-BERT, or using a privacy-preserving scheme that avoids to memorize identifiers PPMLM-BERT) show a similar increase in utility, starting from 0.57 at epoch 0 to 0.72 at epoch 64. At the beginning of the sanitization, except for the repair-only strategy, an erasure step is performed by resetting the weights of some neurons to zero. This reinitialization degrades the utility at the beginning of the sanitization, which then increases as a function of the number of epochs. The results show that the SANI strategy reaches the same level of utility at epoch 77 as at the end of the specialization without protection measures. The pruning strategy, on the other hand, degrades utility slightly. For the repair-only strategy, this corresponds to continuing learning. This is confirmed by an increase in utility in line with that achieved by MLM-BERT at the end of specialization. The sanitization of indirect identifiers from models trained on pseudonymised data shows the same trends. However, the level of utility is slightly lower.

Considering privacy and utility, SANI shows the best trade-off compared to a unlearning strategy based on pruning or repair-only. Repair-only provides the best utility but the worst privacy, and pruning depicts roughly the same privacy than SANI but a smaller utility. The assessment of the sanitization of a fine-tune Causal Language Model (i.e., a GPT-type model) shows the same trend. Due to space constraints, the results are only presented in Appendix A.2.

## 4.3 SANITIZING A PRE-TRAINED MODEL

We now evaluate the ability of SANI to sanitize terms defined as confidential information from a pre-trained Masked Language Model (i.e., a BERT model) without altering its performance on a downstream classification task. Figure 3 shows the evolution of the regurgitation and utility during the sanitization process. Figure 3(a) shows that the off-the-shelf model initially displays a regurgitation level close to 0.25 and that only a single epoch is enough to drastically reduce it close to 0.05. Regurgitation then decreases over the following epochs of sanitization less rapidly. The repair-only strategy, by not resetting the weight of certain neurons, regurgitates more sensitive information than SANI and pruning strategies, which display a similar level.

Figure 3(b), in turn, depicts the utility during sanitization by measuring the performance of a downstream classification task (i.e., detection of the emotion of the text). The results show that the models exhibit similar performance, between 0.92 and 0.93 of F1-scores, throughout the sanitization process. Therefore, sanitizing confidential information has no impact on the model's performance when used in a downstream emotion classification task.

Finally, to better understand this rapid decline in regurgitation after a single epoch, we study which terms are most likely to be regurgitated and at what level during unlearning of SANI. To achieve this goal, Figure 4(a) reports for each term to unlearn their number of repetitions in the training data according to their number of regurgitations observed subsequently. Results clearly show a correlation between the number of times the model was exposed to sensitive terms and the number of regurgitations of these terms, the more repeated, the more regurgitated. Figure 4(b), in turn, depicts the cumulative number of regurgitations according to the number of times the term has been repeated in the training data. Results consistently show that without unlearning (i.e., at epoch 0) the greatest amount of regurgitation occurs for the most frequently repeated sensitive terms during

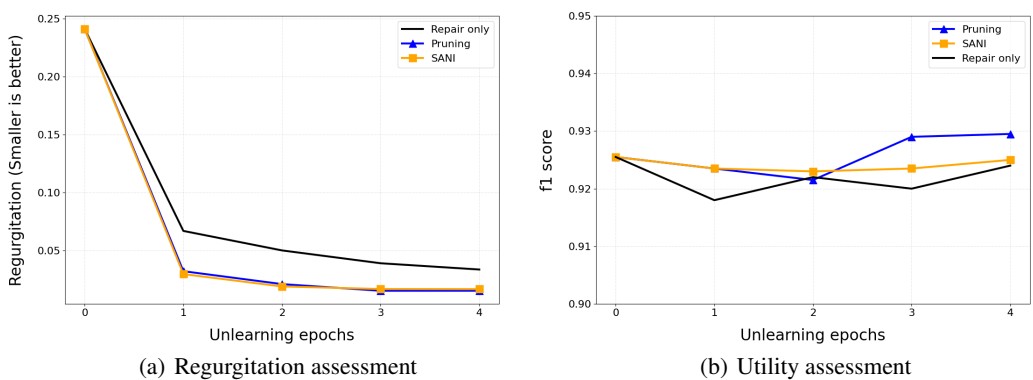

(a) Regurgitation assessment

(b) Utility assessment

Figure 3: Sanitizing a pre-trained BERT model: a single epoch is enough to drastically reduce the regurgitation of confidential information while not affecting the model's performance for a down-stream classification task.

training. After one epoch of unlearning, we observe that the regurgitation associated with the most frequently repeated sensitive information in training decreases drastically. We go from 470,000 regurgitations in total without unlearning, to 60,000 and 30,000 regurgitations after 1 and 2 epochs, respectively. However, counterintuitively, the results show that for very rarely repeated terms in the training data, the number of regurgitations increases slightly after unlearning. This is certainly due to the reduction in the cardinality of the words to be predicted, mechanically increasing the probability of predicting rarely repeated words.

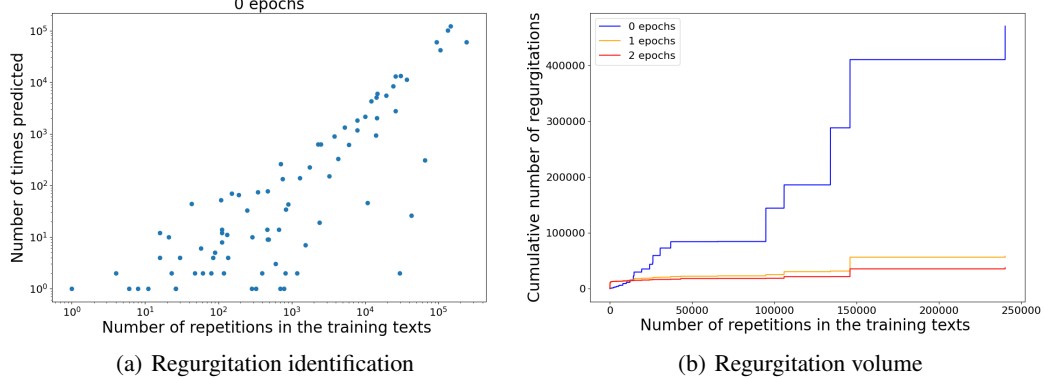

(a) Regurgitation identification

(b) Regurgitation volume

Figure 4: Without unlearning, the most repeated terms in the training data are the most regurgitated, SANI drastically reduces the regurgitation of the most repeated terms.

## 5 CONCLUSION

This article presents SANI, an unlearning strategy to sanitize LLMs, illustrated through two use cases: anonymizing a model (i.e., forgetting direct and indirect identifiers) and removing confidential information from the model. This unlearning strategy can be applied to a model specialized on a corpus of data, or to a pre-trained model. This unlearning strategy addresses the need for model sharing by limiting the risk of subsequent regurgitation of sensitive information, without paying the prohibitive cost of retraining models from scratch. We exhaustively evaluated our approach and demonstrated that it quickly unlearns the sensitive information and offers the best trade-off between the quality of the unlearning and the model performance compared to other baselines. An interesting avenue for future work would be to use and evaluate SANI to remove the influence of biased data and for the removal of backdoors introduced into the model.

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

## A APPENDIX

### A.1 METHODOLOGY

This section provides methodological details useful to reproduce our evaluations. First, when datasets are pseudonymized before being used for model fine-tuning (named MLMA-BERT for BERT and CLMA-GPT for GPT), direct identifiers are replaced by the term "X" in the text. Then, BERT models (i.e., Masked Language Model) were trained by randomly replacing a subset of tokens from the sequence by the <MASK> token, and asking the model to predict them using cross-entropy loss. In our setting, 15% of the words are randomly selected. In the case of PPMLM-BERT, identifying words are excluded from this random selection. If a word chosen to be masked consists of several

tokens, all of them are masked. At each epoch, 15% of non-identifying words are masked. Finally, GPT models (i.e., Causal Language Model) were trained by sequentially (i.e., autoregressively) predicting all tokens. In PPCLM-GPT, if this word is a direct or indirect identifier, it is replaced by a specific padding token that is not taken into account during fine-tuning (i.e., for the computation of the loss).

During the fine-tuning of pre-trained models, all measurements were performed after 4, 8, 16, 32 and 64 epochs to observe their evolution. We used 8 batches of 512 tokens for training. For language modeling, the learning rate starts at 1e-4 and decreases linearly until the end of fine-tuning. For classification, we took the models after 1, 2, 3 and 4 unlearning epochs, and trained them again on a classification task for 4 epochs. The learning rate in these cases starts at 0, 10% of the total number of warmup steps with the learning rate increasing until reaching 2e-5 then decreasing linearly on the remaining 90% steps.

All the computation has been parallelized on a hybrid GPU/CPU computing farm.

## A.2    Sanitizing a fine-tuned Causal Language Model

In this section, we evaluate the ability of SANI to unlearn and remove identifiers (direct and/or indirect) from a Causal Language Model (i.e., a GPT model) specialized with medical data. As in the case of a Masked Language Model (i.e., a BERT model) reported in Section 4.2, we evaluate both the evolution of privacy (i.e., the regurgitation of identifiers) and the model's ability to correctly generate tokens at the right location in the text. The results show the same trends as for BERT sanitization, with a rapid reduction in regurgitation after a single epoch and a better trade-off between unlearning quality and model utility for SANI compared to other baselines.

We however observe a notable difference in the model's utility level, which displays lower performance than for the BERT model (for all baselines). This difference is caused by the absence of identifiers in the context accessed by the model, which have been replaced by padding tokens (see Appendix A.1). Since the model has access to less contextual information when predicting tokens, the quality of the prediction is affected.

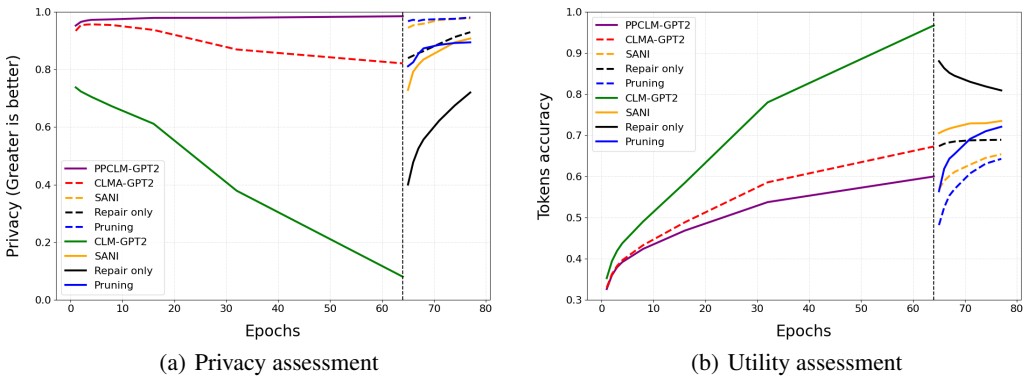

(a) Privacy assessment                         (b) Utility assessment

Figure 5: Sanitizing a fine-tuned GPT model: a single epoch is enough to drastically reduce the regurgitation of identifiers while limiting utility reduction.

