# OpenReview forum: "Leverage Unlearning to Sanitize LLMs"
_ICLR.cc/2026/Conference — Submitted to ICLR 2026_

### Official Review · Reviewer_2TpQ · 2025-10-26

**Soundness:** 2
**Presentation:** 2
**Contribution:** 2
**Rating:** 2
**Confidence:** 4

**Summary:**

This paper proposes SANI, an unlearning-based approach to sanitize large language models by removing memorized sensitive information while preserving utility. Instead of retraining from scratch, SANI first disrupts memorized content by randomly resetting part of the neurons in the final layer, then fine-tunes the model again while excluding sensitive terms so they are not re-learned. The method is evaluated on BERT and GPT models in two practical settings: fine-tuned medical models that must forget patient identifiers, and pre-trained models that must unlearn confidential terms from public datasets. Results show that SANI rapidly reduces regurgitation of sensitive information—often after just one training epoch—while maintaining language modeling and downstream task performance. Compared to pruning and repair-only baselines, it achieves a better trade-off between privacy and utility. The study highlights that frequently repeated training terms are most prone to memorization and are effectively forgotten under SANI, demonstrating its practicality for safe model deployment.

**Strengths:**

1. The paper proposes a simple yet effective erase-and-repair method that does not require full retraining or architectural changes, making it practical for real-world deployment of LLMs that contain sensitive or private data.
2. Extensive experiments on both fine-tuned medical models and pre-trained language models show that SANI significantly reduces sensitive information regurgitation while preserving downstream performance, outperforming pruning and repair-only baselines.

**Weaknesses:**

1. The method resets only the final layer to erase memorized content, but does not provide strong theoretical support for why this layer alone is sufficient to remove deeper internal representations. The approach may fail if sensitive information is stored in earlier layers or attention patterns.
2. SANI primarily focuses on direct regurgitation of exact n-grams. It is unclear whether the method can remove more abstract or paraphrased forms of sensitive knowledge, such as latent identity information or indirect personal attributes.
3. The paper does not test whether the model can relearn forgotten content with minimal fine-tuning, nor does it evaluate resistance to extraction attacks, gradient-based probing, or membership inference attacks after sanitization.
4. All experiments are conducted on BERT-base or GPT-2 small. It is unclear whether the approach remains effective and efficient for instruction-tuned 7B–70B parameter models, where safety and privacy requirements are more critical.

**Questions:**

1. The paper mainly modifies the final layer for memory erasure. How do the authors justify that sensitive information is not embedded in deeper representations or attention patterns?
2. The evaluation of forgetting appears incomplete. The paper mainly tests regurgitation using fixed prompts, but does not examine whether the model still reveals the forgotten information under paraphrased, indirect, or adversarial prompts. In other words, does forgetting remain effective when the prompt is changed?
3. Additionally, it is unclear whether the method has been evaluated on standardized machine unlearning benchmarks such as TOFU [1]. Without such experiments, it is difficult to determine whether the approach generalizes beyond a few manually designed prompts or tasks.

[1] Maini, P., Feng, Z., Schwarzschild, A., Lipton, Z. C., & Kolter, J. Z. (2024). Tofu: A task of fictitious unlearning for llms. arXiv preprint arXiv:2401.06121.

---

### Official Review · Reviewer_Dobv · 2025-10-28

**Soundness:** 2
**Presentation:** 2
**Contribution:** 2
**Rating:** 2
**Confidence:** 4

**Summary:**

This paper proposes SANI, an approach to sanitize large language models by removing memorized sensitive information without retraining from scratch. The method combines a targeted erasure step that randomly resets half of the final-layer neurons to damage fine-grained memorization, followed by a repair/finetuning step that explicitly avoids masking or predicting blacklisted tokens. The authors evaluate SANI on BERT and GPT-2 models using medical records and book datasets, claiming reduced regurgitation of identifiers and confidential content under modest computational cost. Experimental results suggest that one to two unlearning epochs yield substantial privacy improvement with limited utility degradation.

**Strengths:**

The paper tackles a relevant and timely problem: avoiding privacy leakage in pretrained or fine-tuned LLMs prior to model sharing. The approach is conceptually simple and computationally inexpensive, which can be practical for organizations with limited resources. The evaluation includes two architectures (BERT and GPT-2) and two application scenarios, which broadens relevance beyond a single case. The results are easy to interpret since the proposed metrics directly measure regurgitation and utility.

**Weaknesses:**

The technical novelty is limited. The method is essentially a straightforward application of selective last-layer reinitialization combined with masked LM training that excludes sensitive tokens. Both ideas are known, and the paper relies heavily on existing work (e.g., erase-and-repair strategies). The choice of randomly resetting 50 percent of final-layer neurons is not justified, nor is there analysis of which neurons actually encode sensitive content. The evaluation lacks strong baselines such as influence-function-based unlearning, gradient-ascent unlearning, or privacy-neuron localization methods. Regurgitation metrics appear simplistic and not standardized; no membership inference results are provided, despite being a key risk discussed in the paper itself. The datasets used in medical scenarios are extremely small, limiting generalization. Parts of the text contain unclear phrasing and inconsistencies, and figures do not show statistical significance or variance.

**Questions:**

Why do the authors assume that only the last layer stores sensitive memorization? Strong memorization has been shown to exist in attention heads and deeper representation layers. Can you provide supporting analysis?

What motivates the 50% neuron reset ratio? Did you explore more targeted or data-dependent erasure strategies?

How scalable is SANI when applied to models with billions of parameters? Have you evaluated inference-time latency degradation?

How exactly is “regurgitation” defined and detected for generative models? The paper suggests that any prediction of a sensitive token increases the regurgitation count, which seems to penalize predictions even when they are not harmful or not contextually revealing.

Can you provide direct comparison with more competitive state-of-the-art machine unlearning baselines? For example, DF-KD unlearning, influence-theoretic removal, or GA-based LLM unlearning?

The authors discuss membership inference in Section 2 but never evaluate it. Can you include experiments to quantify improvement on MIA resistance?

The repair step seems very similar to simply continuing finetuning while excluding sensitive terms from masking. Under what conditions does the erasure step materially contribute beyond this?

---

### Official Review · Reviewer_SGge · 2025-10-31

**Soundness:** 2
**Presentation:** 2
**Contribution:** 1
**Rating:** 0
**Confidence:** 5

**Summary:**

This paper presents SANI, a practical and efficient unlearning strategy designed to sanitize large language models (LLMs) by removing memorized sensitive information. The authors demonstrate the approach through two key use cases: (1) anonymizing fine-tuned models by unlearning direct and indirect personal identifiers, and (2) sanitizing pre-trained models to eliminate confidential information. SANI operates via a two-phase process—targeted erasure and reconstruction without re-memorization—which enables the model to forget specific information while preserving overall performance. Unlike costly full retraining, SANI achieves this at a fraction of the computational cost. Through extensive empirical evaluations on BERT and GPT models across medical and public datasets, the study shows that SANI drastically reduces regurgitation risks with minimal impact on downstream utility tasks. The method consistently outperforms existing baselines in both privacy preservation and performance retention. The paper concludes with future directions, including extending SANI for bias removal and backdoor mitigation in LLMs.

**Strengths:**

1. Effective and Fast Sanitization: SANI achieves rapid and substantial reduction in sensitive information regurgitation after just 1–2 unlearning epochs.

2. Strong Utility-Privacy Trade-off: It maintains downstream task performance (e.g., word prediction, classification) while improving privacy, demonstrating minimal utility loss even after sanitization.

3. Versatility Across Use Cases: SANI works for both fine-tuned (e.g., medical data) and pre-trained models (e.g., public corpora with confidential terms), showing broad applicability.

**Weaknesses:**

1. Limited methodological novelty. The paper presents SANI as a new unlearning method, but it lacks theoretical depth or clear innovation. The main idea of resetting part of the model and then fine-tuning it while avoiding certain tokens is similar to existing erasure and repair approaches. There are no formal definitions, equations, or theoretical analysis to help evaluate how or why the method works.

2. Insufficient baseline comparisons. The evaluation does not include enough recent or strong baselines. Many newer unlearning methods exist, such as those based on influence functions, certified removal, or parameter-efficient techniques. The paper only compares with pruning, repair only, and a simple retraining setup, which makes its performance claims less convincing.

3. Evaluation limited to small models. The experiments use only BERT base and GPT2 small. There is no evidence that the method works on larger or more widely used models such as GPTJ, LLaMA, or T5. This limits the paper’s relevance for real-world applications.

**Questions:**

see Weaknesses.

---

### Meta-Review · Area_Chair_UcxR · 2026-01-06

**Summary:**

The authors propose SANI, an unlearning approach designed to sanitize LLMs by removing memorized sensitive information. The method operates through two phases: an erasure phase that randomly resets neurons in the final layers to disrupt fine-grained memorization, followed by a repair phase that fine-tunes the model while avoiding re-memorization of sensitive tokens. While all the reviewers acknowledged that the paper addresses a relevant and timely problem, they found the work lacks methodological novelty. In particular, the core idea of resetting part of the model and then fine-tuning resembles existing erase-and-repair strategies. The authors evaluated their proposed algorithm on two different architectures and application scenarios, and show a reasonable trade-off between privacy preservation and performance retention compared to the simple baselines included in the paper.

Overall, the paper received unanimously negative scores from all three reviewers. While the problem setting is relevant and the approach is simple, the lack of methodological novelty, insufficient baselines, limited model scale, and incomplete evaluation limits the contribution of the work. I recommend rejection and encourage the authors to address the reviewer concerns in a future submission. Best of luck with the next steps!

**Reviewer Concerns:**

The authors did not engage in a discussion with the reviewers and did not submit a rebuttal.

**Reviewer Scores:**

Not applicable as the authors did not submit a rebuttal.

---

### Decision · Program_Chairs · 2026-01-26

Reject